# Highly efficient synthesis of non-planar macrocycles possessing intriguing self-assembling behaviors and ethene/ethyne capture properties

Lijun Mao[1], Yang Hu[1], Qian Tu[1], Wei-Ling Jiang[1], Xiao-Li Zhao[1], Wenjing Wang[2], Daqiang Yuan [2], Jin Wen[3] & Xueliang Shi [1✉]

It has been a challenging topic and perpetual task to design and synthesize covalent macrocycles with characteristic self-assembling behaviors and excellent host-guest properties in supramolecular chemistry. Herein, we present a family of macrocyclic diphenylamine[n] arenes (**DPA[n]s**, n = 3–7) consisting of methyldiphenylamine units through a facile one-pot synthesis strategy. Unlike many other reported macrocyclic arenes, the resultant non-planar **DPA[n]s** feature intrinsic π-π stacking interactions, interesting self-assembling behaviors and ethene/ethyne capture properties. Specifically, strong multiple intermolecular edge-to-face aromatic interactions in **DPA[3]** have been systematically investigated both in solid and solution states. The intriguing findings on the intermolecular edge-to-face stacking interaction mode in the macrocycle would further highlight the importance of noncovalent π-π interaction in supramolecular self-assembly. This study will also shed light on the macrocyclic and supramolecular chemistry and, we expect, will provide a direction for design and synthesis of covalent macrocycles in this area.

[1] Shanghai Key Laboratory of Green Chemistry and Chemical Processes, School of Chemistry and Molecular Engineering, East China Normal University, 3663N. Zhongshan Road, 200062 Shanghai, People's Republic of China. [2] State Key Laboratory of Structural Chemistry, Fujian Institute of Research on the Structure of Matter, Chinese Academy of Sciences, 350002 Fujian, Fuzhou, People's Republic of China. [3] Institute of Organic Chemistry and Biochemistry, Academy of Sciences of the Czech Republic, 16610 Prague 6, Czech Republic. ✉email: xlshi@chem.ecnu.edu.cn

Macrocycles represent one of the most intriguing molecular entities that have been playing an essential role in the establishment and development of supramolecular chemistry. Design and synthesis of macrocycles featuring characteristic host-guest properties and self-assembling behaviors are always one of the cutting-edge research topics in the supramolecular chemistry[1–5]. The development of macrocyclic host-guest system is not only very conducive to exploring, explaining and understanding the non-covalent interactions but also endows the macrocycles with more recognition functions[6–18]. Meanwhile, the self-assembly of macrocycles into higher-order nanostructures such as columns and channels has found wide applications in the fields of supramolecular delivery systems and porous materials[19–21]. Normally, the intermolecular H-bonding and π–π stacking (often parallel stacking) interactions are the main driving forces employed to create such channel type macrocycles (Fig. 1a). For example, several well-developed macrocycles including cyclic peptides[22–25], phenylacetylene macrocycles[26–28], metal-organic coordination macrocycles[29–31], aromatic oligoamide macrocycles[32–34], bis-urea macrocycles[21,35], etc. feature channel type self-assembling behavior and have demonstrated potential applications for the artificial transmembrane ion channels, separation, gas storage, and catalyst[36,37] (Fig. 1c). Therefore, it is of high interest to efficiently prepare macrocycles possessing characteristic self-assembling behaviors, excellent host-guest properties, as well as some intriguing intermolecular non-covalent interactions.

Macrocyclic arenes, as one kind of important macrocycles, have always attracted much attention mainly on account of their easy preparation, convenient functionalization, interesting host-guest interaction, and enormous potential application in varieties of functional materials[1–5,38]. The self-assembly of macrocyclic arenes into columns or channels is believed to be an efficient approach to implement their practical applications in supramolecular delivery systems and porous materials[36,37]. However, not all the reported macrocyclic arenes seem to be able to stack into columns or channels. For example, without post-modification, some representative macrocyclic arenes, such as calix[n]arenes, pillar[n]arenes, corona[n]arenes and so on, do not favor channel-type molecular packing mainly because of their non-planar structural conformation or the lack of the robust intermolecular non-covalent forces[39]. Thus, it is of great challenge to construct macrocyclic arenes exhibiting unconventional self-assembling behaviors, as well as distinct recognition functions.

Under above premise, we herein present a family of macrocyclic diphenylamine[n]arenes (**DPA[n]s**, $n = 3$–7) consisting of methyldiphenylamine units, with alternative methylene and nitrogen bridges (Fig. 2). The idea of deliberately choosing methyldiphenylamine as building block is based on the following considerations: (a) the electron rich nature of N-substituted diphenylamine ensures that the dynamic Friedel-Crafts (FC) alkylation reaction can only occur at the *para* position, which is useful for the macrocycle formation; (b) the macrocycles with methylene and nitrogen bridges will display the well-tuned cavity and intrinsic conformation, which may produce interesting solid-state packing motif and self-assembling behavior; (c) lone pairs on nitrogen can participate in resonance and increase the electron density of **DPA[n]s** and consequently enhance the binding affinity towards the guest molecules. Thus, in this study, a series of **DPA[n]s** ($n = 3$–7) were successfully constructed via one-pot condensation reaction. Interestingly, it is found that the resultant non-planar **DPA[3]** and **DPA[4]** could maintain a pseudo three-dimensional cavity, as well as intrinsic π–π stacking interactions and interestingly unconventional self-assembling behaviors. Particularly, ideally and strong multiple intermolecular edge-to-face aromatic interactions have been observed in the crystal of **DPA[3]** (Fig. 1b). Consequently, the robust edge-to-face aromatic

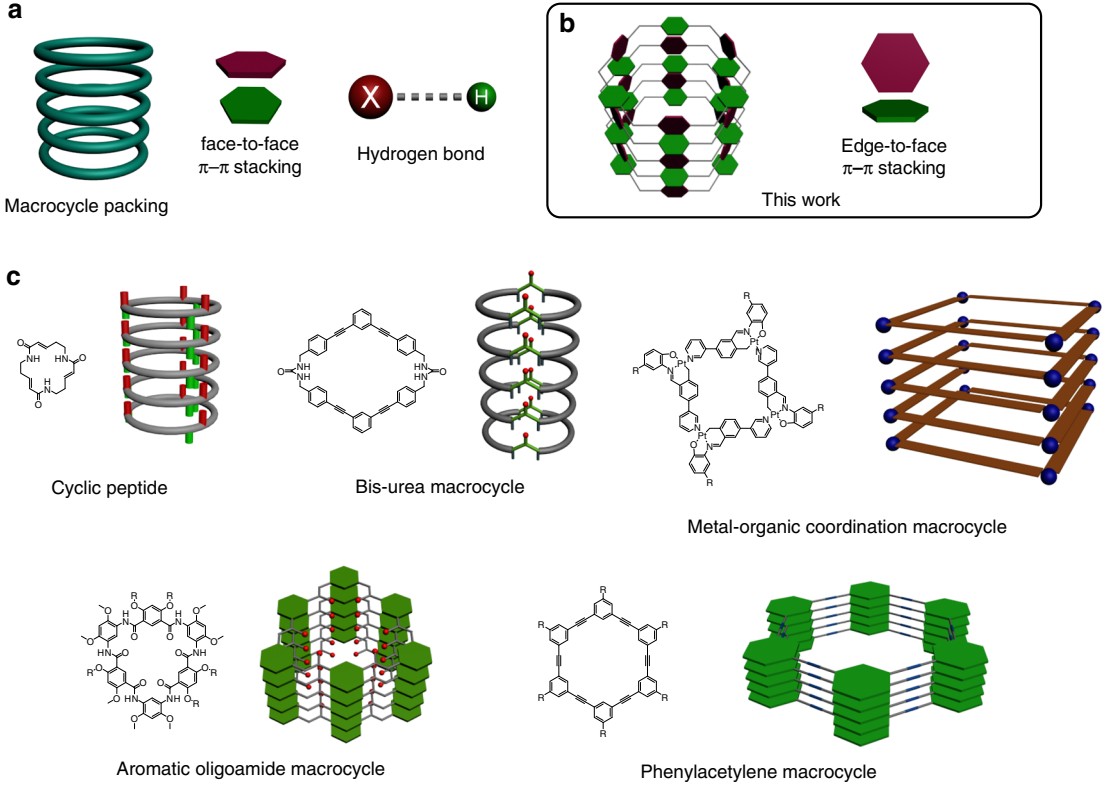

**Fig. 1 Different packing of macrocycles. a** Illustration of the molecular packing mode of macrocycle and two main driving forces (face-to-face π–π stacking and hydrogen bond) of the packing. **b** The packing of **DPA[n]s** studied in this article with edge-to-face π–π stacking. **c** The packing mode of cyclic peptide, bis-urea macrocycle, metal-organic coordination macrocycle, aromatic oligoamide macrocycle, and phenylacetylene macrocycle.

interactions facilitate its columnar self-assembling behavior which is elucidated by the indefinite isodesmic model. Moreover, **DPA[3]** with a guest accessible channel has displayed hosting ability to capture the guests like ethene and ethyne gases, indicating its tremendous potential in gas separation and adsorption in the future. To the best of our knowledge, there are few macrocyclic arenes capable of self-assembling into columns via the purely intermolecular edge-to-face aromatic interactions. Beside, **DPA[3]** capable of hosting both ethene and ethyne gases, which is directly confirmed by X-ray crystallography in this work, also represents a very rare host material in the area of gas storage and separation.

## Results

**Synthesis and characterization of DPA[n]s (n = 3–7).** The synthetic route to **DPA[n]s** is depicted in Fig. 2. The synthetic strategy is very similar to that of the reported pillar[n]arenes and calix[4]pyrroles, where a FC alkylation reaction is applied. The substrate N-methyldiphenylamine containing nitrogen atom facilitates the FC alkylation reaction that proceeds at the para-position, which is useful for the macrocycle formation. The effects of catalyst, solvent, reaction time and temperature on the yield of the product have been thoroughly investigated, and the synthetic reaction conditions were optimized (Supplementary Fig. 1 and Table 1). It was found that the catalyst is a pacing factor for the successful synthesis of the macrocycle, and FeCl₃·6H₂O catalyst leads to a better reproducibility and higher yield formation of the macrocycles. Moreover, it also revealed that only condensation reaction in dichloromethane or 1,2-dichloroethane could afford the desired **DPA[n]s**, which might be linked to the reaction mechanism (e.g., the stability of the reaction intermediate) or the solubility of the product **DPA[n]s**. Finally, a series of **DPA[n]s** including **DPA[3]** (yield 20%), **DPA[4]** (yield 10%), **DPA[5]** (yield 1%), **DPA[6]** (yield 0.5%), and **DPA[7]** (yield 0.3%) were prepared when the reaction was carried out in dichloromethane at room temperature for 6 h (Supplementary Fig. 1). Notably, the yield of **DPA[3]** could be significantly improved to 55% upon

increasing the reaction time to 12 h. Moreover, with the change of the solvent from dichloromethane to 1,2-dichloroethane, the yield of **DPA[3]** could be further improved to 75%, which might be ascribed to the template effect of the solvent. The substituent effect of N-substituted diphenylamine on the macrocycle formation was also surveyed. The FC alkylation reaction of substrates bearing benzyl and methyl acetate groups afforded **DPA[3]-a** and **DPA[3]-b**, respectively, which are the derivatives of **DPA[3]**, in good yield (Supplementary Table 1). In contrast, the preliminary results demonstrated that triphenylamine derivatives were unable to form macrocyclic arenes on this condition (Supplementary Table 1) probably because of the competing Scholl reaction[40]. All **DPA[n]s** were fully characterized by ¹H NMR, ¹³C NMR, and mass spectrometry (Supplementary Figs. 56–76). In particular, **DPA[3]**, **DPA[4]**, **DPA[3]-a**, and **DPA[3]-b** were unambiguously confirmed by X-ray crystallographic analysis.

**Crystal structures of DPA[n].** Single crystals suitable for X-ray crystallographic analysis were obtained for **DPA[3]** and **DPA[4]** by slow diffusion of methanol into chloroform solution (Fig. 3). In this study, **DPA[n]s** have even-numbered unsubstituted arenes and adopt non-planar conformation. All arene rings were found to alternately twist with two set of bridge angle, where the angle of nitrogen bridge is bigger than that of methylene bridge, which obviously differed from pillar[n]arenes composed of alkoxy substituted arenes and sole methylene bridge[38,41]. Consequently, **DPA[3]** exhibited a pseudo cylindrical configuration with three-dimensional cavity, where six benzene rings aligned alternately perpendicular and parallel to the macrocycle plane (Fig. 3a, b). In contrast, **DPA[4]** curved and adopted a saddle-shape conformation in the solid state (Fig. 3c, d).

Similar to many macrocyclic arenes[39,41], **DPA[3]** also possessed a racemic mixture of enantiomers in its single crystal (Fig. 3b) attributed to the different orientation of the Ar-rings coupled to their non-symmetrical substitution which is similar to what has

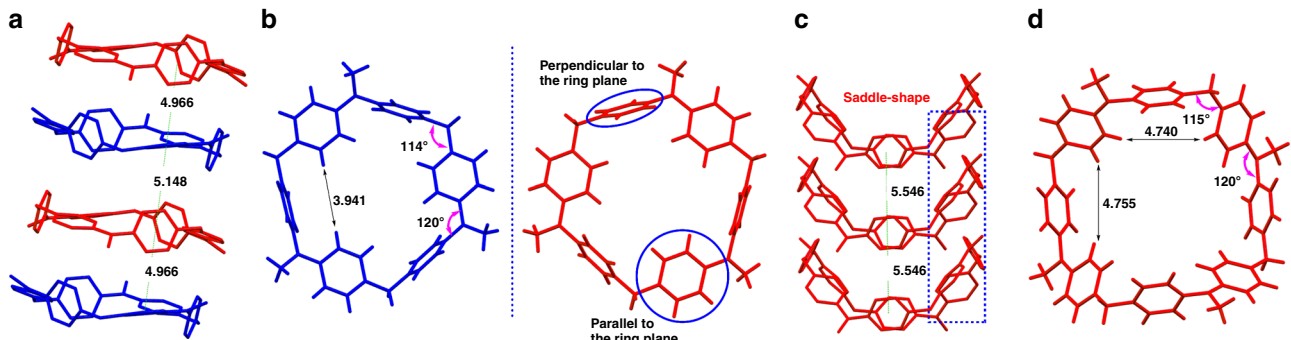

**Fig. 2 One-step synthesis of DPA[n]s (n = 3–7).** Synthesis of **DPA[n]s** through an iron-catalyzed Friedel–Crafts reaction under room temperature. Solvent: dichloromethane or 1,2-dichloroethane.

**a** **b** **c** **d**

**Fig. 3 X-ray crystallographic structures.** Stacking model of **DPA[3]** (**a**) and **DPA[4]** (**c**), and single crystal structures of **DPA[3]** (**b**) and **DPA[4]** (**d**).

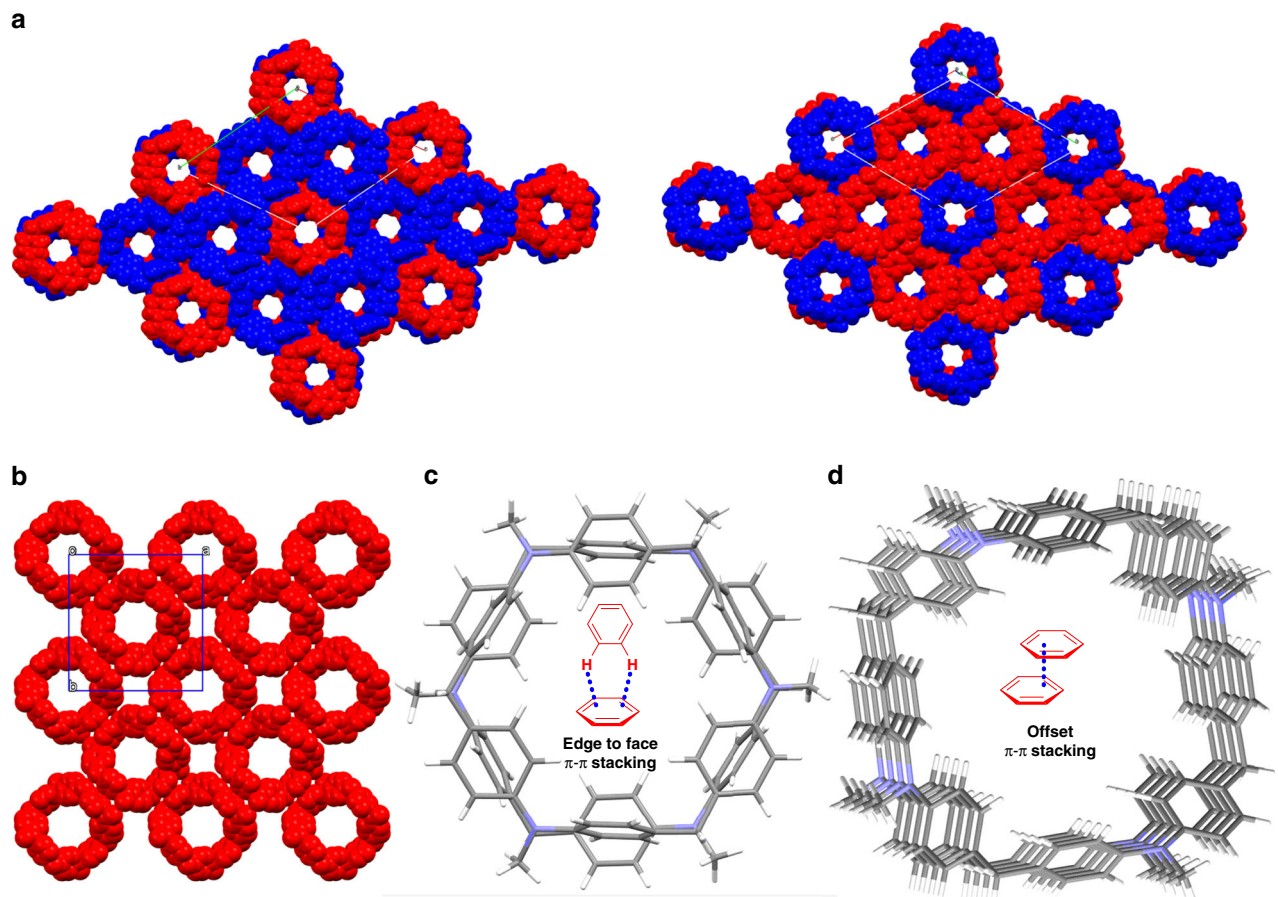

**Fig. 4 Single crystal packing model.** Crystal packing of **DPA[3]** (**a**) and **DPA[4]** (**b**) and intermolecular interaction of **DPA[3]** (edge to face π–π stacking, **c**) and **DPA[4]** (offset π–π stacking, **d**).

been observed in pillar[n]arenes[38] and cycloparaphenylene[6]. The column separation of the two enantiomers was not applicable because of their rapid racemization at ambient temperature. As a consequence, we calculated the rotation barriers of **DPA[3]** whose transition state structure was obtained by the synchronous transit-guided Quasi-Newton (STQN) method with the QST3 option[42,43], and the calculation results confirmed that the rapid interconversion of **DPA[3]** enantiomers for the free enthalpy change as small as 1.23 kcal/mol (Supplementary Fig. 2). What is more, the variable-temperature [1]H NMR experiment likewise suggested that **DPA[3]** underwent rapid racemization even at very low temperature, which is caused from the flipping motion of the Ar-rings[44] (Supplementary Figs. 3 and 4). By contrast, no enantiomer was found in **DPA[4]** because of the existence of a symmetry center (Fig. 3d and Supplementary Fig. 50). Therefore, the incorporation of nitrogen atom brings a significant impact on the conformation, cavity, electron density and even chirality of **DPA[n]s**.

With regard to the crystal packing, both two molecules were aligned into hollow column with infinite array, though their structural conformations were non-planar. Figure 4a illustrates the front and back view of two layers of the crystal packing in the ab-plane of **DPA[3]**. **DPA[3]** was aligned in a hexagonal packing perpendicular to the a–b plane, thereinto each layer containing two enantiomers with 1:1 ratio aligned alternatively along the c axis. Correspondingly, **DPA[4]** displayed a square arrangement that is also perpendicular to the a-b plane (Fig. 4b). Notably, these two macrocycles can be stacked to form continuous channels similar to cyclic peptides, phenylacetylene macrocycles, metal-organic

coordination macrocycle, aromatic oligoamide, bis-urea macrocycles, etc., while the two macrocycles featuring distinct intermolecular aromatic π–π interactions. More specifically, an ideal edge-to-face π–π stacking geometry in **DPA[3]** was observed with a centroid-to-centroid distance of ~5 Å (Figs. 3a and 4c). Instead, offset π–π stacking geometry dominated the packing in **DPA[4]** with a longer centroid-to-centroid distance of ~5.5 Å (Figs. 3c and 4d). In contrast, ordered and multiple intermolecular π–π stacking interaction was rarely observed in the conventional non-planar macrocyclic arenes, e.g., pillararenes[38,41], calixarene[45–48], and cycloparaphenylene[6,49], since their non-planar molecular conformations do not favor the intermolecular packing.

At the same time, single crystals suitable for X-ray crystallographic analysis were also obtained for **DPA[3]-a** and **DPA[3]-b** by slow diffusion of methanol into chloroform solution (Supplementary Figs. 53–55). Interestingly, besides the skeleton similarities, the edge to face π–π stacking motif and columnar self-assembling behaviors are also observed in the crystal of **DPA[3]-a** and **DPA[3]-b** in spite of their different substituents on nitrogen atom compared with **DPA[3]**. This finding implied that such edge to face π–π stacking might be an inherent property of **DPA[3]** and its derivatives. Consequently, the robustly intrinsic edge to face π–π interactions are the primary driving force to induce the columnar self-assembly of non-planar **DPA[3]**.

**DFT calculation on DPA[n]**. All attempts to grow single crystals for the other **DPA[n]s** (n = 5–7) homologous have been unsuccessful. So, a computational prediction of their conformations was

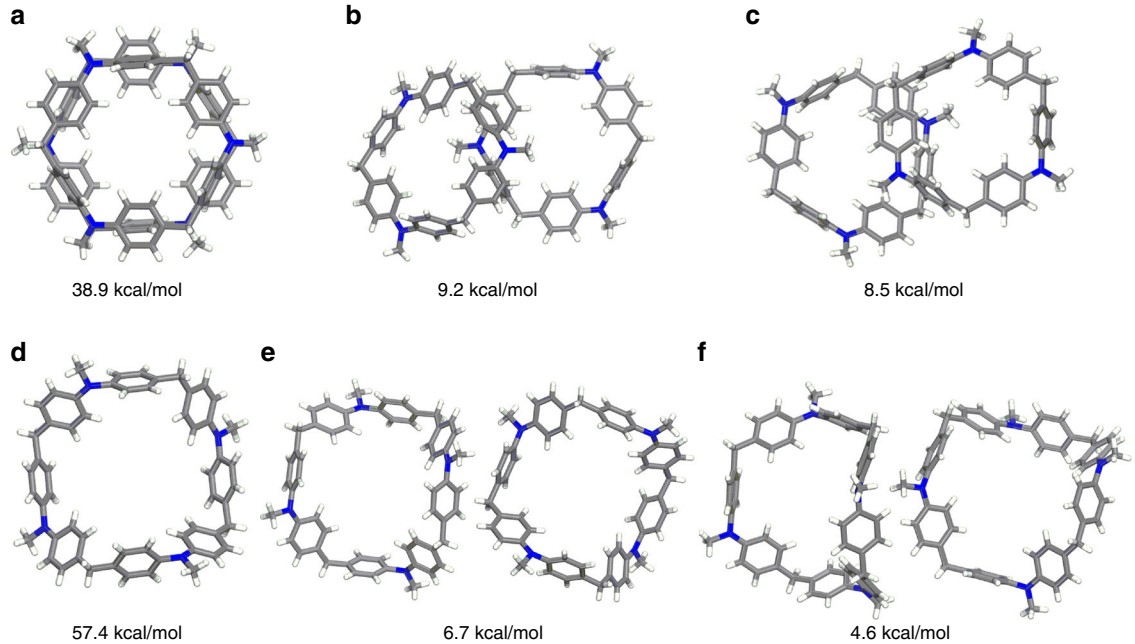

**a** 38.9 kcal/mol  **b** 9.2 kcal/mol  **c** 8.5 kcal/mol

**d** 57.4 kcal/mol  **e** 6.7 kcal/mol  **f** 4.6 kcal/mol

**Fig. 5 Binding energies of macrocycles.** Binding energies of different dimeric packing arrangements for **DPA[3]** (**a**–**c**) and **DPA[4]** (**d**–**f**) obtained from DFT calculation based on B97D3/cc-pVDZ level.

performed at B3LYP/cc-pVDZ level, which implied that **DPA[n]s** displayed size-dependent conformations in the solid state (Supplementary Fig. 5). The calculated conformations of **DPA[3]** and **DPA[4]** were well consistent with the results obtained from their X-ray crystallographic analysis, indicating the chosen calculated method herein was appropriate and correct. The calculated molecular geometries of the high-order **DPA[n]s** ($n = 5$–$7$) revealed that the macrocycles became further distorted with the increase of their size. Besides, the calculated HOMO electron density on the nitrogen bridge is notably higher than that on the methylene bridge, indicating that the lone pairs on nitrogen could likely participate in resonance and increase the electron density of **DPA[n]s** (Supplementary Fig. 6).

For the purpose of better understanding the molecular packing, DFT calculation based on B97D3/cc-pVDZ level was carried out to compare the binding energies of both **DPA[3]** and **DPA[4]** dimer, wherein their experimental dimer observed from single crystal packing and a few theoretically predicted packing dimers were calculated and compared. The binding energies of the selected dimers for **DPA[3]** and **DPA[4]** were listed in Fig. 5, Supplementary Tables 2 and 3. It should be noted that the theoretically predicted dimerized structures showed much lower binding energies than that of the dimer observed in single crystal, which indicated that the experimental packing is more stable than the theoretically predicted packing arrangements. The large binding energies of the dimers suggest the robustly intrinsic π–π stacking interactions in **DPA[3]** and **DPA[4]**, which would facilitate their columnar self-assembling behavior. Therefore, these systems can be regarded as self-assembled nanotubes, in which **DPA[n]** units are held together by the intrinsic π–π stacking interactions.

**Aggregation behavior of DPA[n]s.** π–π stacking interaction as one kind of important weak non-covalent interactions refers to specific intermolecular attraction involving the π orbitals of aromatic rings. It is well-known that π–π stacking interactions are of high importance to the stability of secondary structure in nucleic acid polymers such as DNA[50]. Stacking interactions are also one of the most widely used driving forces for the self-assembly in supramolecular chemistry[51,52]. Several π–π stacking geometric

configurations including face-to-face parallel stacking, offset parallel stacking, T-shaped edge-to-face stacking, and tilted T-shaped edge-to-face stacking have been observed in the crystals of macrocyclic arenes[53,54]. Among them, the parallel-stacked geometry is the most common configuration that dominates in the self-assembly of most planar aromatic molecules and macrocycles such as aforementioned phenylacetylene macrocycles, metal-organic coordination macrocycles, aromatic oligoamides, etc. Comparatively, the pure edge-to-face stacking modes in the self-assembly of macrocycles have never been investigated. Actually, edge-to-face interaction is energetically attractive about 1.5 kcal/mol and it is the most stable aromatic stacking of π–π interaction between neutral molecules[53,54]. However, ideal edge-to-face aromatic interactions are disfavored in solution, which can only be observed in solid state and are sensitive to temperature due to the restricted internal mobility[53,54]. Therefore, there is a high demand for the design of molecule possessing intrinsic intermolecular edge-to-face aromatic interactions. In view of the aforementioned crystallographic analysis and DFT calculation results, we envision that **DPA[3]** with the ideal edge-to-face π–π stacking geometry (Fig. 4c) should exhibit different self-assembling behavior compared with **DPA[4]** that possessed the offset π–π stacking geometry (Fig. 4d). And these findings direct our focus to study their self-assembly mechanism and behavior in detail.

The π–π stacking interactions of aromatic molecules usually produce additional ring-current effects. Consequently, their [1]H NMR spectra are sensitive to concentration change, and the stacking of aromatic molecules results in upfield shifts of the protons. Thus, concentration-dependent [1]H NMR spectroscopy as a method of choice could be employed to determine the strength of the self-assembly (association constant K) of our system[55–57]. As expected, the aromatic chemical shifts of **DPA[n]s** were very dependent on their concentrations rather than temperature, leading to distinct upfield shifts with the increase of concentration (Fig. 6a, Supplementary Figs. 9 and 12). Considering the unique packing motif of **DPA[n]s**, we here choose the indefinite isodesmic model (the EK model), which assumes that the addition of a molecule to a stack occurs with the same equilibrium constant $K_E$ as other molecules[55] (Fig. 6b). The constants $K_E$ in this model were obtained by using

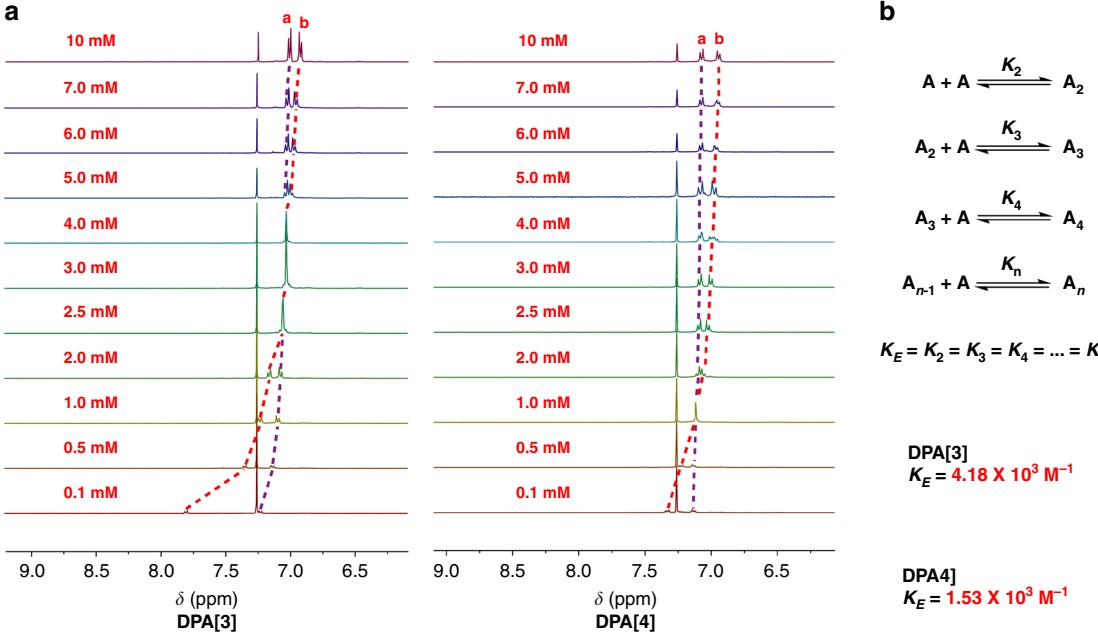

**Fig. 6 Partial ¹H NMR spectra and equilibrium constant of macrocycles. a** Partial stacked ¹H NMR spectra (400 MHz, 298 K) of **DPA[3]** (left) and **DPA[4]** (right) from 0.1 mM to 10 mM in CDCl₃. **b** Step aggregation of macrocycles and equilibrium constant of **DPA[3]** and **DPA[4]**.

nonlinear least-squares fitting (Fig. 6b, Supplementary Figs. 10 and 13), and the $K_E$ value as high as $4.18 \times 10^3$ and $1.53 \times 10^3$ M⁻¹ were determined for **DPA[3]** and **DPA[4]**, respectively[57]. Remarkably, the association constants for both two **DPA[n]s** are much higher than those of stacked planar macrocycles[50,56], implying the considerable driving force for self-assembly in this study. The different $K_E$ values of **DPA[3]** and **DPA[4]** are likely to be associated with their different π–π stacking geometries, i.e., the slightly bigger $K_E$ of **DPA[3]** can be attributed to the edge-to-face π–π interaction that is more stable than offset π–π interaction[53,54].

The distribution of the mole fraction of each oligomer[57] for both **DPA[3]** and **DPA[4]** is also determined and shown in Fig. 7. As illustrated, the great majority of oligomers is present in the form of monomer under low concentration (0.1 mM) for both macrocycles. With the increase of concentration, the mole fractions of the higher stacked species ($n > 10$) rise dramatically, indicating their concentration-dependent self-assembling behavior. Besides, the data also revealed that, under the same concentration, the fractions of the high-order oligomers in **DPA[3]** are significantly more than those of **DPA[4]**, which is in accordance with their unequal constants $K_E$ and different π–π stacking geometries. Because of the exponential relationship between the mole fraction and $K_E$, small changes in $K_E$ value could result in relatively large changes in the mole fraction. Thus, the above qualitative and quantitative analytics disclosed that the robustly intrinsic π–π interactions are the primary driving force to induce the columnar self-assembly of non-planar **DPA[n]s**, markedly different from the conventional non-planar macrocyclic arenes. And also multiple intermolecular edge-to-face aromatic interactions in a macrocycle have been systematically investigated, and demonstrated that such interactions can function as the main driving forces for the macrocycle self-assembly.

Their self-assembling behavior was further studied by scanning electron microscope (SEM), atomic force microscope (AFM), 2-D diffusion-ordered spectroscopy (DOSY) and dynamic light scattering (DLS) experiments. The aggregates of both macrocycles formed under low concentration (0.1 mM) displayed an irregular clumpy morphology visualized by SEM (Supplementary Fig. 33) and AFM (Supplementary Figs. 31 and 32). In contrast, long flexible nanorod

with uniform morphology can be observed at high concentration (1.0 mM) (Fig. 8a, b) similar to those of the most reported columnar assembled macrocycles[25,29,33]. The DOSY experiments of various concentrations of **DPA[3]** and **DPA[4]** in CDCl₃ indicated that the diffusion coefficients $D$ values of **DPA[3]** and **DPA[4]** progressively decreased from $6.31 \times 10^{-9}$ to $1.00 \times 10^{-9}$ m² s⁻¹ and $7.94 \times 10^{-9}$ to $2.00 \times 10^{-9}$ m² s⁻¹, respectively, with the increase of the concentration from 1.0 mM to 100 mM (Supplementary Figs. 15–30). In addition, in the same concentration, $D$ values of **DPA[3]** are smaller than those of **DPA[4]**. Since the rate of diffusion is inversely proportional to the hydrodynamic size, the increase of the hydrodynamic size of macrocycles with the increase of their concentrations might provide the further support for their self-assembling behaviors in the solution state. In addition, DLS data revealed that the hydrodynamic diameter of **DPA[3]** and **DPA[4]** increased gradually from 0.674 to 413 nm and 0.655 to 10.1 nm, respectively, with the increase of the macrocycles concentration from 0.001 to 1.00 mM in CHCl₃ (Fig. 8c, d). Meanwhile, under the same concentration, the aggregates of **DPA[3]** displayed a relatively bigger hydrodynamic diameter than that of **DPA[4]**. Thus, all these data further supported the different concentration-dependent self-assembling behaviors of **DPA[3]** and **DPA[4]** in the solution state.

**Host-guest properties of DPA[n]s.** In view of the unique packing motif of **DPA[3]** with intrinsic edge-to-face π–π stacking interactions, well-defined cavity size in the solid state and accessible channels, we assume that **DPA[3]** could form host-guest complex with some guests containing π bonds such as ethene and ethyne. Notably, it has been very rare that the capture of ethene and ethyne by means of covalent macrocycles due to the lack of the binding sites[58,59]. As a consequence, the ethene and ethyne capture ability of **DPA[3]** was initially checked through thermogravimetric analysis (TGA). From the TGA curve of **DPA[3]** powder that was immersed in ethene atmosphere for 48 h, a ~4% weight loss was observed when temperature raised above 200 °C (Supplementary Fig. 35), while **DPA[3]** is stable under 400 °C (Supplementary Fig. 34). Similarly, the TGA curve of

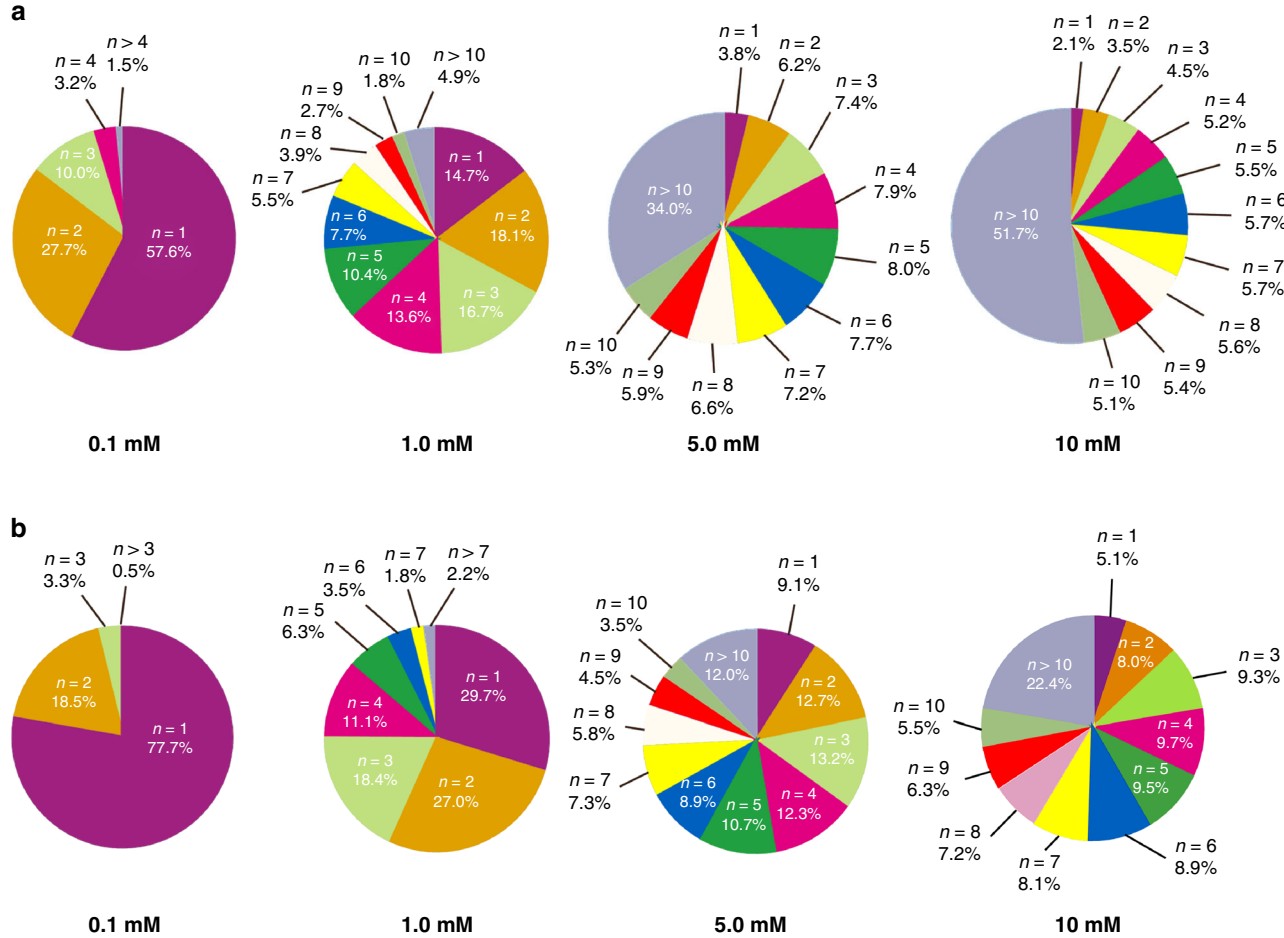

**Fig. 7 Populations of oligomeric n-mers in terms of mass fractions of DPA[n].** Mass fractions of **DPA[3]** (**a**) and **DPA[4]** (**b**) at 0.1, 1.0, 5.0, and 10 mM.

the ethyne-immersed **DPA[3]** powder exhibited about 3% weight loss when the temperature raised above 120 °C (Supplementary Fig. 36). The results implied that one **DPA[3]** may bind about one ethene or ethyne molecule, and resulting in the host-guest complex of **DPA[3]**⊃ethene and **DPA[3]**⊃ethyne. In addition, $^1$H NMR spectrum of the host-guest complexes exhibited the typical signal of ethene and ethyne proton at 5.40 and 1.91 ppm, respectively (Fig. 9a, b). The integration of the ethene and ethyne protons by NMR also supported the plausible gas capture ratio. What's more, the NOESY spectra of the complexes revealed through-space NOEs between the internal aromatic proton ($H_b$) of **DPA[3]** and guest molecules' protons of ethene and ethyne, while no through-space NOEs between the external methyl proton ($H_d$) and guest molecules' proton was observed (Fig. 9c, d), which further implied that the guest molecules are likely to be absorbed in the cavity of macrocycle. Fortunately, we successfully obtained the single crystal of **DPA[3]**⊃ethene and **DPA[3]**⊃ethyne suitable for X-ray diffraction (Fig. 9e–h). From the single crystal, the ethene and ethyne molecule actually located in the cavity of **DPA[3]**, which is in agreement with the NOESY results. However, the interaction between **DPA[3]** and guest molecules is uncertain because ethene and ethyne exhibit disorder in the crystal of **DPA[3]**⊃ethene and **DPA[3]**⊃ethyne. DFT calculations implied that **DPA[3]** favors binding to π-bonded ethene and ethyne rather than ethane which is a saturated hydrocarbon (Supplementary Fig. 39). In addition, $^1$H NMR investigations of host-guest complexes between **DPA[3]** and a series of small molecules indicated the existence of weak host-guest interaction between **DPA[3]** and alkenes and alkynes such as n-hexylene, n-hexyne, styrene and phenylacetylene

(Supplementary Figs. 40–47). Thus, we infer that the formation of host-guest complex might be attributed to the weak CH-π or electrostatic interaction between **DPA[3]** and guest[60–62]. To the best of our knowledge, this is a very rare example to demonstrate that small covalent macrocycle is capable of hosting ethene and ethyne gas directly confirmed by X-ray crystallography[63]. It is likely that the inherent intermolecular edge-to-face aromatic interactions and unique channel type self-assembling behavior in **DPA[3]** are essential to facilitate the host-guest properties toward ethene and ethyne. The preliminary results also highlight that **DPA[3]** as porous material has potential applications of gas separation and adsorption in the future[19,20].

## Discussion

In summary, a series of non-planar **DPA[n]s** ($n = 3$–7) were successfully synthesized, and their structures and properties were well studied. Unlike many other reported macrocyclic arenes, the resultant non-planar **DPA[n]s** feature intrinsic π–π stacking interactions, interesting self-assembling behaviors and ethene/ethyne capture properties. Specifically, **DPA[3]** and **DPA[4]** could self-assemble into columns with well-defined channel on account of their robust π–π stacking interactions. Notably, strong multiple intermolecular edge-to-face aromatic interactions in **DPA[3]** have been systematically investigated both in solid and solution states. And the preliminary results also demonstrated that the edge-to-face aromatic interactions can function as the main driving force for the macrocycle self-assembly. Benefitting from the three-dimensional cavity, the inherent intermolecular edge-to-face aromatic interactions and columnar self-assembling behavior, the

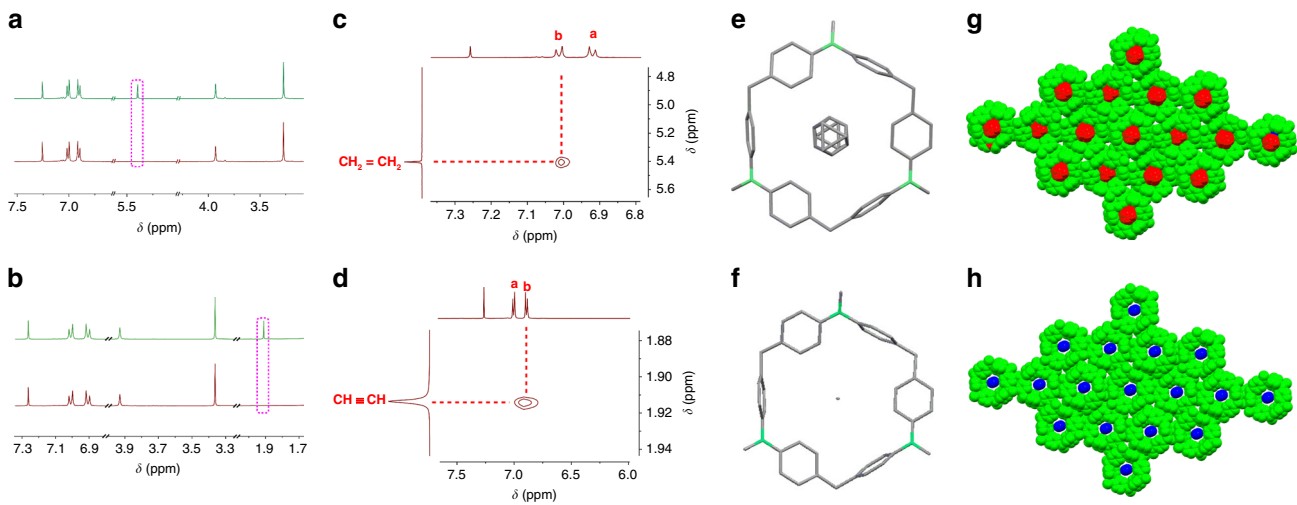

**Fig. 8 SEM and DLS of DPA[n].** SEM images of **DPA[3]** (**a**) and **DPA[4]** (**b**) at a high concentration (1.0 mM) and DLS spectra of DPA[3] (**c**) and **DPA[4]** (**d**) in CHCl$_3$.

**Fig. 9 NMR spectra and crystal of host-guest complex.** Partial stacked $^1$H NMR spectra (400 MHz, 298 K) of **DPA[3]** before and after immersing in ethene (**a**) and ethyne (**b**) atmosphere for 48 h. Partial 2D NOESY (500 MHz, 298 K) of **DPA[3]**⊃ethene (**c**) and **DPA[3]**⊃ethyne (**d**) in CDCl$_3$. X-ray crystal structure and 3D packing of **DPA[3]**⊃ethene (**e**, **g**) and **DPA[3]**⊃ethyne (**f**, **h**).

designed **DPA[3]** demonstrated its encapsulation toward ethene and ethyne molecules, thus implying its potential applications of gas separation and adsorption in the future. This study provides a direction for the design and self-assembly of functional non-planar macrocyclic arenes. Our work also sheds light on the fundamental mechanism of multiple intermolecular edge-to-face π–π stacking in macrocycles, that helps to better understand the importance of π–π stacking interaction in supramolecular self-assembly.

## Methods

All solvents were dried according to the standard procedures and all of them were degassed under $N_2$ for 30 min before use. All air-sensitive reactions were carried out under inert $N_2$ atmosphere. [1]H and [13]C NMR spectra were recorded at 400 MHz with a Mercury plus 400 spectrometer at 298 K and tetramethylsilane (TMS) as an internal reference. The [1]H and [13]C NMR chemical shifts are reported relative to the residual solvent signals. Coupling constants ($J$) are denoted in Hz and chemical shifts (δ) in ppm. Multiplicities are denoted as follows: s = singlet, d = doublet. 2D NMR spectra (NOESY and DOSY) and variable-temperature [1]H NMR spectra were recorded on Bruker 500 MHz Spectrometer. Mass spectra were recorded with Thermo Scientific LTQ XL spectrometer with methanol or acetonitrile as solvents. For the single crystals, the data sets were treated with the SQUEEZE program to remove highly disordered solvent molecules. The crystallographic formulae include the number of solvent molecules was suggested by the SQUEEZE program. Scanning electron microscopy (SEM) was performed on a Hitachi S-4800 microscope. The AFM samples were prepared by drop casting method using mica sheet as substrate. All the AFM images were obtained on a Dimension FastScan (Bruker), using ScanAsyst mode under ambient condition.

## Data availability

The data that support the findings of this study are available from the authors on reasonable request, see author contributions for specific data sets. The X-ray crystallographic coordinates for structures reported in this study have been deposited at the Cambridge Crystallographic Data Centre (CCDC), under deposition numbers CCDC 1882113 (**DPA[3]**), 1948482 (**DPA[4]**), 1948483 (**DPA[3]⊃ethene**), 1975012 (**DPA[3]⊃ethyne**), 2015861 (**DPA[3]-a**), and 2011992 (**DPA[3]-b**). These data can be obtained free of charge from The Cambridge Crystallographic Data Centre via www.ccdc.cam.ac.uk/data_request/cif.

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

## Acknowledgements
X.S. acknowledges the financial supports sponsored by Shanghai Sailing Program (19YF1412900) and the Fundamental Research Funds for the Central Universities. J.W. thanks to Austrian Science Fund (FWF): M 2709-N28. We thank Yan-Fei Niu, Tan Ji, Yi Qin, Ji-Chuang Shen, Wei-Jian Li, and Xi Liu for their help in collecting experimental data. We thank the staffs from BL17B beamline of National Facility for Protein Science in Shanghai (NFPS) at Shanghai Synchrotron Radiation Facility, for assistance during data collection.

## Author contributions
X.S. and L.M. conceived the project, analyzed the data, and wrote the manuscript. L.M. performed the most of experiments. X.-L.Z. conducted single crystal analyses. J.W. carried out DFT calculation. Y.H., Q.T., W.-L.J., W.W., and D.Y. helped in experiments and data analyses. All authors discussed the results and commented on the manuscript.

## Competing interests
The authors declare no competing interests.
