## [Peer Review File · Nature Communications]

REVIEWER COMMENTS

Reviewer #1 (Remarks to the Author):

This manuscript deals with the synthesis of a new class of macrocycles, named as “diphenylamine[n]arenes” (DPA[n]s), which have been obtained by a FeCl₃-catalyzed Friedel-Crafts (FC) alkylation of methyldiphenylamine with formaldehyde. The resulting macrorings are non-planar, show interesting self-assembling behaviors and ethene/ethyne capture properties. The DPA[n]s are an interesting addition to the continuously growing field of new emerging macrocycles which are expanding the supramolecular properties of the more classical calixarene and resorcinarene hosts. Therefore, I believe that this work is of interest for the general audience of Nature Communications and consequently I support its publication in this journal providing that the following minor points are appropriately addressed by the authors.

- 1) Title + text: Probably it's better to say “self-assembling behaviors” instead of “self-assembly behaviors”.
- 2) Line 9. A rewording of “...methyldiphenylamine based macrocyclic diphenylamine[n]arenes...” is necessary.
- 3) Line 10. “cyclic arenes” should be changed to “macrocyclic arenes”. All the text should be checked (e.g. L178, L273, L338).
- 4) L78. Delete “particularly for those of arene-based macrocycles”, is an useless repetition.
- 5) L79. “some represented macrocyclic arenes” should be “some representative macrocyclic arenes”.
- 6) L86. The new name “diphenylamine[n]arenes” is only mentioned in the abstract. It cannot be traced into the text. Therefore, it should be at least eventually mentioned here (see below).
- 7) L86. Of course, the new name “diphenylamine[n]arenes” could be substituted with “aza-pillararenes”. It appears that the authors have already considered this option as demonstrated by the name “N-pillar[n]arene” used in Figure 6b. This inconsistencies should be solved. “Aza-pillararenes” seems to be preferable.
- 8) L89. Delete “kinetic”.
- 9) L90. Change “...which is the prerequisite to form the macrocycles...” to “...which is useful for the macrocycle formation....”.
- 10) L104. A rewording of “...to capture the guests of ethene and ethyne gases ...” is necessary.
- 11) L107. Should be “...capable of hosting both ethene and ethyne gases...”.
- 12) L115. See point #9.
- 13) L118. Reword “...the kind of catalyst is a pacing factor...”.
- 14) L120. Reword “...has a better repeatable and higher yield...”.
- 15) L147. Should be “..adopted a saddle-shape conformation in the solid state...”.
- 16) L152. The nitrogen is not “intrinsically chiral”. Actually, it is achiral. Here, the chirality is due to the different orientation of the Ar rings coupled to their non-symmetrical substitution. It is very similar to what has been observed with pillararenes and CPP. Modify.
- 17) L156. A ref. is necessary for “the synchronous transit-guided Quasi-Newton (STQN) method with the QST3 option”.
- 18) L157. Should be “rapid interconversion”.
- 19) L158. The Supplementary Figure 2 should be redrawn by keeping the nitrogens fixed in their

places.

- 20) L159. The VT NMR in the Supp. Info do not give much information because the reached temperature is not low enough. VT NMR in CH₂Cl₂ at a lower temperature should be more useful.
- 21) L160. This racemization is quite different with respect "to the nitrogen inversion process". The nitrogen here is almost planar. This racemization is due to the flipping motion of the Ar-rings (see point #16).
- 22) L162. I don't see the "symmetry center" in the DPA[4] molecule. Here, there is no inversion center and no symmetry planes: DPA[4] molecules are also chiral. Modify.
- 23) L175. Should be (Figs. 3a and 4c). See also L177.
- 24) Figures 3 and 4. The difference between blue and green color is difficult to see. Change it.
- 25) Figure 4. The "alpha" and "beta" symbols are not explained.
- 26) L185. Should be "DFT calculation on DPA[n]s".
- 27) L188. Should be "size-dependent conformations in the solid state".
- 28) L202. Reword "It should be noted that all theoretically predicted dimerized structures showed the lower binding energies".
- 29) L213. Should be "Aggregation behavior of DPAs".
- 30) L214. Should be "intermolecular attraction involving the π orbitals of aromatic rings".
- 31) L221. Should be "crystals of macrocyclic arenes".
- 32) L231. Should be "molecules possessing intrinsic".
- 33) L241. Should be "employed to determine the strength".
- 34) L272. Should be "non-planar".
- 35) L296. Should be "Host-guest properties of DPAs".
- 36) L335 and L355. There are two "Discussion" sections.
- 37) L343. Reword "in solid and solution states for the first time. And the preliminary results also demonstrated".
- 38) L347. Reword "molecular recognition capability to ethene and ethyne molecules".

Reviewer #2 (Remarks to the Author):

This paper describes first the synthesis of a new family of p(cyclophane)-related macrocycles, along with the optimised preparation of one of them. The second part of the manuscript deals with their structure in the solid state, and corresponding inclusion properties. In this later part of the manuscript, ethene and ethyne gases are used as guests. An exploration of their aggregation behaviour in solution is also disclosed.

Regarding this later point, it is not clear if the structures observed by MEB are present in solution, or the simple result of an evaporation/crystallisation effect during the preparation of the sample.

Despite the fact that this work appears as well conducted and described, it doesn't seem to show a sufficient level of novelty and originality compared with the plethora of related studies to deserve

publication in Nature Communications. The new organisation observed for the aromatics in the solid state (edge to face, possibly resulting in the observed inclusion of ethene and ethyne gases, the main result of the paper), is interesting, but does not represent a breakthrough in the field. Moreover, the relationship between the observed edge to face packing in the solid state and the observed inclusion effects is possible, but not demonstrated.

However, this work is interesting, (as is the direct observation of included C₂H₂ and C₂H₄ by ¹H NMR spectroscopy), and thus deserves publication in large audience journals, such as Chemistry, a European Journal, or Communications Chemistry.

REVIEWER COMMENTS

Reviewer #1 (Remarks to the Author):

This manuscript deals with the synthesis of a new class of macrocycles, named as “diphenylamine[n]arenes” (DPA[n]s), which have been obtained by a FeCl₃-catalyzed Friedel-Crafts (FC) alkylation of methyldiphenylamine with formaldehyde. The resulting macrorings are non-planar, show interesting self-assembling behaviors and ethene/ethyne capture properties. The DPA[n]s are an interesting addition to the continuously growing field of new emerging macrocycles which are expanding the supramolecular properties of the more classical calixarene and resorcinarene hosts. Therefore, I believe that this work is of interest for the general audience of Nature Communications and consequently I support its publication in this journal providing that the following minor points are appropriately addressed by the authors.

Reply: We greatly appreciate the reviewer’s positive comments on the work in this manuscript.

1) Title + text: Probably it’s better to say “self-assembling behaviors” instead of “self-assembly behaviors”.

Reply: The descriptions of “self-assembly behaviors” have all been changed to “self-assembling behaviors” in the context.

2) Line 9. A rewording of “...methyldiphenylamine based macrocyclic diphenylamine[n]arenes...” is necessary.

Reply: The sentence in Line 9 was reorganized: “Herein, we present a new family of macrocyclic diphenylamine[n]arenes (DPA[n]s, n = 3-7) consisting of methyldiphenylamine units through a facile one-pot synthesis strategy.”

3) Line 10. “cyclic arenes” should be changed to “macrocyclic arenes”. All the text should be checked (e.g. L178, L273, L338).

Reply: Based on the reviewer's advice, all the words of "cyclic arenes" have been changed to "macrocyclic arenes".

4) L78. Delete "particularly for those of arene-based macrocycles", is an useless repetition.

Reply: Following the reviewer's advice, the statement of "particularly for those of arene-based macrocycles" has been deleted.

5) L79. "some represented macrocyclic arenes" should be "some representative macrocyclic arenes".

Reply: Based on the reviewer's advice, "represented" has been changed to "representative".

6) L86. The new name "diphenylamine[n]arenes" is only mentioned in the abstract. It cannot be traced into the text. Therefore, it should be at least eventually mentioned here (see below).

Reply: Following the reviewer's advice, the name of "diphenylamine[n]arenes" has been re-mentioned in the sentence of "Under above premise, we herein present a new family of macrocyclic "phenylamine[n]arenes (**DPA[n]s**, $n = 3-7$) consisting of methyldiphenylamine units, with alternative methylene and nitrogen bridges (Fig. 2)." in Introduction section.

7) L86. Of course, the new name "diphenylamine[n]arenes" could be substituted with "aza-pillararenes". It appears that the authors have already considered this option as demonstrated by the name "N-pillar[n]arene" used in Figure 6b. This inconsistencies should be solved. "Aza-pillararenes" seems to be preferable.

Reply: We do agree with the reviewer that the macrocyclic arenes presented in this manuscript may bear some resemblance to the currently fashionable macrocyclic arenes of "Pillar[n]arenes" in supramolecular chemistry. Some comparisons between these two kinds of macrocycles have also been discussed in the manuscript. We did

consider choosing the name of “aza-pillararenes” or “N-substituted pillararenes” when the authors generated the first draft of the manuscript. However, we were worried that this may be confusing to some of those reading this manuscript because the macrocycles herein exhibited some different properties regarding their structure, packing motif, self-assembling behaviors and host-guest properties compared to conventional pillar[n]arenes. Thus, we finally used “diphenylamine[n]arenes” to name such series macrocycles.

8) L89. Delete “kinetic”.

Reply: Following the reviewer’s advice, “kinetic” was deleted.

9) L90. Change “...which is the prerequisite to form the macrocycles...” to “...which is useful for the macrocycle formation...”.

Reply: Following the reviewer’s advice, the sentence of “...which is the prerequisite to form the macrocycles...” has been changed to “...which is useful for the macrocycle formation...”.

10) L104. A rewording of “...to capture the guests of ethene and ethyne gases ...” is necessary.

Reply: Following the reviewer’s advice, the description of “...to capture the guests of ethene and ethyne gases ...” has been changed to “...to capture guests like ethene and ethyne gases ...”.

11) L107. Should be “...capable of hosting both ethene and ethyne gases...”.

Reply: Following the reviewer’s advice, the description of “...capable of simultaneously hosting ethene and ethyne gas...” has been changed to “...capable of hosting both ethene and ethyne gases...”.

12) L115. See point #9.

Reply: Following the reviewer’s advice, we have made the corresponding changes.

13) L118. Reword "...the kind of catalyst is a pacing factor...".

Reply: Following the reviewer's advice, the description of "...the kind of catalyst is a pacing factor..." has been changed to "the catalyst is a pacing factor...".

14) L120. Reword "...has a better repeatable and higher yield...".

Reply: Following the reviewer's advice, the description of "...has a better repeatable and higher yield..." has been changed to "...and FeCl₃•6H₂O catalyst leads to a better reproducibility and higher yield formation of the macrocycles."

15) L147. Should be "...adopted a saddle-shape conformation in the solid state...".

Reply: Following the reviewer's advice, the description of "...adopted a saddle-shape configuration in its solid state..." has been changed to "...adopted a saddle-shape conformation in the solid state...".

16) L152. The nitrogen is not "intrinsically chiral". Actually, it is achiral. Here, the chirality is due to the different orientation of the Ar rings coupled to their non-symmetrical substitution. It is very similar to what has been observed with pillarenes and CPP. Modify.

Reply: The authors agree with the reviewer's comment about the chirality of **DPA[3]**. We have made the corresponding changes in the chirality discussion part in the manuscript, and a related literature has been cited (Org. Lett. 2011, 13, 9, 2480-2483):
"*...What is more, the variable-temperature ¹H NMR experiment likewise suggested that DPA[3] underwent rapid racemization even at very low temperature, which is caused from the flipping motion of the Ar-rings⁴⁴ (Supplementary Figures 3 and 4)....*"

17) L156. A ref. is necessary for "the synchronous transit-guided Quasi-Newton (STQN) method with the QST3 option".

Reply: Following the reviewer's advice, reference 42 and 43 have been cite about

STQN method with the QST3 option.

18) L157. Should be “rapid interconversion”.

Reply: Following the reviewer’s advice, “rapid transformation” has been changed to “rapid interconversion”.

19) L158. The Supplementary Figure 2 should be redrawn by keeping the nitrogens fixed in their places.

Reply: Following the reviewer’s advice, the Supplementary Figure 2 was replaced.

20) L159. The VT NMR in the Supp. Info do not give much information because the reached temperature is not low enough. VT NMR in CH₂Cl₂ at a lower temperature should be more useful.

Reply: Following the reviewer’s advice, the VT NMR of **DPA[3]** in CD₂Cl₂ was re-recorded and the Supplementary Figure 3 was modified. The temperature was reached about -70 °C, which is the lowest temperature that our NMR machine can reach. As shown in Figure R1, the VT NMR results indicated that **DPA[3]** underwent rapid racemization even at very low temperature, which is caused from the flipping motion of the Ar-rings.

Figure R1. Partial stacked variable-temperature ^1H NMR (500 MHz) spectra of **DPA[3]** (10 mM) in CD_2Cl_2 .

21) L160. This racemization is quite different with respect “to the nitrogen inversion process”. The nitrogen here is almost planar. This racemization is due to the flipping motion of the Ar-rings (see point #16).

Reply: The authors agree with the reviewer’s comment about the chirality of **DPA[3]**. We have made the corresponding changes in the chirality discussion part in the manuscript, and a related literature has also been cited (Org. Lett. 2011, 13, 9, 2480-2483):

*“...What is more, the variable-temperature ^1H NMR experiment likewise suggested that **DPA[3]** underwent rapid racemization even at very low temperature, which is caused from the flipping motion of the Ar-rings⁴⁴ (Supplementary Figures 3 and 4)....”*

22) L162. I don’t see the “symmetry center” in the **DPA[4]** molecule. Here, there is no inversion center and no symmetry planes: **DPA[4]** molecules are also chiral. Modify.

Reply: The authors have carefully checked the crystal structure of **DPA[4]** molecule,

and the symmetry centre of **DPA[4]** was illustrated in Figure R2.

Figure R2. The symmetry centre of **DPA[4]**.

23) L175. Should be (Figs. 3a and 4c). See also L177.

Reply: "Figs. 3a and 4c" and "Figs. 3c and 4d " have replaced previous ones.

24) Figures 3 and 4. The difference between blue and green color is difficult to see. Change it.

Reply: Following the reviewer's advice, Figures 3 and 4 have been modified.

25) Figure 4. The "alpha" and "beta" symbols are not explained.

Reply: The "alpha" and "beta" symbols indicated the front and back view of two layers of the crystal packing in the ab-plane of **DPA[3]**. The "alpha" and "beta" symbols have been deleted in Figure 4 to avoid the possible misunderstanding.

26) L185. Should be "DFT calculation on **DPA[n]s**".

Reply: "DFT calculation of **DPA[n]s**" has been changed to "DFT calculation on **DPA[n]s**".

27) L188. Should be "size-dependent conformations in the solid state".

Reply: Following the reviewer's advice, "size-dependent conformations in their solid

structures” has been changed to “size-dependent conformations in the solid state”.

28) L202. Reword “It should be noted that all theoretically predicted dimerized structures showed the lower binding energies”.

Reply: Following the reviewer’s advice, we have reworded the sentence:

“It should be noted that the theoretically predicted dimerized structures showed much lower binding energies than that of the dimer observed in single crystal, which indicated the experimental packing is more stable than the theoretically predicted packing arrangements.”

29) L213. Should be “Aggregation behavior of DPAs”.

Reply: “Aggregation behavior of macrocycle” has been changed to “Aggregation behavior of **DPA[n]s**”.

30) L214. Should be “intermolecular attraction involving the π orbitals of aromatic rings”.

Reply: Following the reviewer’s advice, “intermolecular attraction between aromatic rings containing π orbitals” has been changed to “intermolecular attraction involving the π orbitals of aromatic rings”.

31) L221. Should be “crystals of macrocyclic arenes”.

Reply: Following the reviewer’s advice, “crystals of aromatic arenes” has been changed to “crystals of macrocyclic arenes”.

32) L231. Should be “molecules possessing intrinsic”.

Reply: We have deleted “that” in the sentence.

33) L241. Should be “employed to determine the strength”.

Reply: The word “quantificationally” has been deleted.

34) L272. Should be “non-planar”.

Reply: We have replaced the wrong word with “non-planar”.

35) L296. Should be “Host-guest properties of DPAs”.

Reply: We have replaced “Host-guest interaction of macrocycle” with “Host-guest properties of **DPA[n]s**”.

36) L335 and L355. There are two “Discussion” sections.

Reply: The second "Discussion" has been changed to "Additional information".

37) L343. Reword “in solid and solution states for the first time. And the preliminary results also demonstrated”.

Reply: Following the reviewer’s advice, we have reworded the sentence:

“Notably, strong multiple intermolecular edge-to-face aromatic interactions in DPA[3] have been systematically investigated for the first time both in solid and solution states.”

38) L347. Reword “molecular recognition capability to ethene and ethyne molecules”.

Reply: Following the reviewer’s advice, we have reworded the sentence:

“...the designed **DPA[3]** demonstrated its encapsulation toward ethene and ethyne molecules...”

Reviewer #2 (Remarks to the Author):

This paper describes first the synthesis of a new family of p(cyclophane)-related macrocycles, along with the optimised preparation of one of them. The second part of the manuscript deals with their structure in the solid state, and corresponding inclusion properties. In this later part of the manuscript, ethene and ethyne gases are used as guests. An exploration of their aggregation behaviour in solution is also disclosed.

Reply: We greatly appreciate the reviewer's positive comments on the work in this manuscript.

Regarding this later point, it is not clear if the structures observed by MEB are present in solution, or the simple result of an evaporation/crystallisation effect during the preparation of the sample.

Reply: The authors do agree with the reviewer's comment that the self-assembling aggregates observed by MEB or crystallography may not be present in solution. Indeed, it is a scientific interest to investigate the different self-assembling/aggregation behaviors of organic molecules in solution and solid states. The aggregation of organic molecules in the solid state can be precisely characterized by various microscopy techniques (SEM, TEM, AFM and so on). Moreover, we can directly observe their self-assembling behaviors through X-ray crystallography. In contrast, the aggregates of molecules are very difficult to be directly visualized in solution state even through the advanced characterization methods/facilities. Considering the reviewer's concern about the aggregation behaviour, particularly in the solution state, the authors have supplemented two experiments to demonstrate the intrinsic edge to face π - π stacking motif and columnar self-assembling behaviors of **DPA[3]**.

First, in order to demonstrate the intrinsic edge to face π - π stacking motif of **DPA[3]**, the derivatives of **DPA[3]**, namely **DPA[3]-a** and **DPA[3]-b** bearing benzyl and methyl acetate group, respectively, have also been synthesized. As shown in Figure

R3, the edge to face π - π stacking motif and columnar self-assembling behaviors are also observed in the crystal of **DPA[3]-a** and **DPA[3]-b** in spite of their different substituents on nitrogen atom. This finding might further support the authors' statement that the robustly intrinsic edge to face π - π interactions are the primary driving force to induce the columnar self-assembly of non-planar **DPA[3]**.

Figure R3. The edge to face π - π stacking motif and columnar self-assembling behaviors of (a) **DPA[3]-a** and (b) **DPA[3]-b**.

Some discussions have been presented in the revised manuscript:

“At the same time, single crystals suitable for X-ray crystallographic analysis were also obtained for DPA[3]-a and DPA[3]-b by slow diffusion of methanol into chloroform solution (Supplementary Figure 54). Interestingly, besides the skeleton similarities, the edge to face π - π stacking motif and columnar self-assembling behaviors are also observed in the crystal of DPA[3]-a and DPA[3]-b in spite of their different substituents on nitrogen atom compared with DPA[3]. This finding implied that such edge to face π - π stacking might be an inherent property of DPA[3] and its derivatives. Consequently, the robustly intrinsic edge to face π - π interactions are the primary driving force to induce the columnar self-assembly of non-planar DPA[3].”

In addition, 2D diffusion-ordered spectroscopy (DOSY) was employed to investigate the self-assembling behaviors of **DPA[3]** and **DPA[4]** in the solution state. Generally, the rate of diffusion is inversely related to the molecular weight/size, *i.e.*, the diffusion coefficients (D) is inversely proportional to the hydrodynamic size. As shown in Figure R4-R18, D values of **DPA[3]** and **DPA[4]** obtained from DOSY experiments progressively decreased with the increase of their concentrations from 1.0 mM to 100 mM. The decrease of the hydrodynamic size of macrocycles with the increase of their concentrations further supported the self-assembling behaviors of **DPA[3]** and **DPA[4]**. Meanwhile, diffusion coefficients of **DPA[3]** were smaller than those of **DPA[4]** when they were in the same concentration, which is also consistent with their unequal equilibrium constant K_E . In summary, the information obtained by concentration-dependent ^1H NMR spectroscopy, scanning electron microscope (SEM), atomic force microscope (AFM), 2-D diffusion-ordered spectroscopy (DOSY), dynamic light scattering (DLS) experiments, together with X-ray crystallography are all consistent and support the columnar self-assembling behaviors of **DPA[3]** and **DPA[4]** in both solution and solid states.

Figure R4. DOSY (500 MHz, 298 K) of 1.0 mM **DPA[3]** in CDCl_3 .

Figure R5. DOSY (500 MHz, 298 K) of 2.0 mM **DPA[3]** in CDCl_3 .

Figure R6. DOSY (500 MHz, 298 K) of 5.0 mM **DPA[3]** in CDCl_3 .

Figure R7. DOSY (500 MHz, 298 K) of 10 mM **DPA[3]** in CDCl_3 .

Figure R8. DOSY (500 MHz, 298 K) of 20 mM **DPA[3]** in CDCl_3 .

Figure R9. DOSY (500 MHz, 298 K) of 50 mM **DPA[3]** in CDCl_3 .

Figure R10. DOSY (500 MHz, 298 K) of 100 mM **DPA[3]** in CDCl_3 .

Figure R11. DOSY (500 MHz, 298 K) of 1.0 mM **DPA[4]** in CDCl_3 .

Figure 12. DOSY (500 MHz, 298 K) of 2.0 mM **DPA[4]** in CDCl_3 .

Figure 13. DOSY (500 MHz, 298 K) of 5.0 mM DPA[4] in CDCl₃.

Figure 14. DOSY (500 MHz, 298 K) of 10 mM DPA[4] in CDCl₃.

Figure 15. DOSY (500 MHz, 298 K) of 20 mM DPA[4] in CDCl₃.

Figure 16. DOSY (500 MHz, 298 K) of 50 mM DPA[4] in CDCl₃.

Figure 17. DOSY (500 MHz, 298 K) of 100 mM DPA[4] in CDCl₃.

Figure R18. Concentration dependence of diffusion coefficient D (500 MHz, 298 K,

CDCl₃) in the solution of **DPA[3]** and **DPA[4]**.

Some discussions have been presented in the revised manuscript:

“The DOSY experiments of various concentrations of DPA[3] and DPA[4] in CDCl₃ indicated that the diffusion coefficients D values of DPA[3] and DPA[4] progressively decreased from 6.31×10^{-9} to $1.00 \times 10^{-9} \text{ m}^2 \text{ s}^{-1}$ and 7.94×10^{-9} to $2.00 \times 10^{-9} \text{ m}^2 \text{ s}^{-1}$, respectively, with the increase of the concentration from 1.0 mM to 100 mM (Supplementary Figures 15-30). In addition, in the same concentration, D values of DPA[3] are smaller than those of DPA[4]. Since the rate of diffusion is inversely proportional to the hydrodynamic size, the decrease of the hydrodynamic size of macrocycles with the increase of their concentrations might provide the further support for their self-assembling behaviors in the solution state.”

Despite the fact that this work appears as well conducted and described, it doesn't seem to show a sufficient level of novelty and originality compared with the plethora of related studies to deserve publication in Nature Communications.

Reply: Regarding the novelty and originality of this work that the referee concerned, this full *Article* included a lot of original works from molecular design and synthesis to various structural and physical characterizations, and to theoretical calculations/discussion. In this manuscript, the authors presented a family of novel non-planar macrocyclic arenes of **DPA[3]** and **DPA[4]** exhibiting intrinsic π - π stacking interactions, interestingly unconventional self-assembling behaviors and host-guest properties with ethene and ethyne molecules. The keywords of this manuscript, such as macrocyclic chemistry, aromatic π - π interactions, self-assembly and host-guest chemistry are of general interest to those working in supramolecular chemistry, organic chemistry, theoretic chemistry, and even material science. For example, aromatic π - π interactions have demonstrated to have relevant importance in chemistry and biology. Several π - π stacking geometric configurations including face-to-face parallel stacking, offset parallel stacking, T-shaped edge-to-face stacking, and tilted T-shaped edge-to-face stacking have been observed in the crystals of macrocyclic arenes. Among them, edge-to-face stacking has been the least studied

probably because ideal edge-to-face aromatic interactions are disfavored in solution, which can only be observed in solid state (Acc. Chem. Res. 2001, 34, 885-894; Chem. Sci., 2012, 3, 2191-2201). Besides, the host-guest chemistry/inclusion of ethene and ethyne might receive broad interests from both the academia and the industry. To the best of our knowledge, it has been very rare that a covalent macrocycle can capture both ethene and ethyne due to the lack of the binding sites.

In summary, the authors would like to highlight the main findings of this study:

- 1) This work might represent a rare case of macrocycle that possesses intrinsic edge to face π - π stacking in both solution and solid states.
- 2) The multiple intermolecular edge-to-face aromatic interactions have been systematically investigated in both solution and solid states for the first time, and demonstrated that such interactions can function as the main driving forces for the macrocycle self-assembly.
- 3) Moreover, this work also represents a rare case of macrocyclic arene that can absorb ethene and ethyne gases, which is demonstrated by TGA, NMR and X-ray crystallography.

The new organisation observed for the aromatics in the solid state (edge to face, possibly resulting in the observed inclusion of ethene and ethyne gases, the main result of the paper), is interesting, but does not represent a breakthrough in the field. Moreover, the relationship between the observed edge to face packing in the solid state and the observed inclusion effects is possible, but not demonstrated.

Reply: The authors fully understand the reviewer's concern about the underlying relationship between the observed edge to face packing of **DPA[3]** and its host-guest interaction with ethene and ethyne gases. Therefore, some additional experiments have been carried out to understand the relationship. The results implied that **DPA[3]** may interact with ethene and ethyne gases through the potential weak CH/ π interactions or electrostatic interaction.

As stated in the manuscript, the interaction between **DPA[3]** and guest molecules is uncertain because ethene and ethyne exhibit disorder in the crystal of **DPA[3]**⊃ethene and **DPA[3]**⊃ethyne. The authors speculated that the existence of weak CH/ π

interactions or electrostatic interaction between **DPA[3]** and π -bonded ethene and ethyne gases might account for their host-guest interaction (J. Phys. Chem. A 2007, 111, 753-758; Phys. Chem. Chem. Phys. 2007, 9, 1680-1687; Chem. Phys. Lett. 2013, 557, 59-65). The energetic profiles of **DPA[3]** complexes with ethane, ethene, and ethyne, respectively, were carried out the single point-energy calculations at the B3LYP/6-31G (Figure R19). The positive binding energy implied that the host-guest interaction between **DPA[3]** and ethane is unfavorable probably because ethane is a saturated hydrocarbon without π -bond that could not interact with **DPA[3]**.

Figure R19. Binding energies of host-guest complexes between **DPA[3]** and ethane, ethene, and ethyne obtained from DFT calculation based on B3LYP/6-31G level.

The host-guest interaction between **DPA[3]** and a series of (potential) guest molecules including *n*-hexane, *n*-hexylene, *n*-hexyne, ethylbenzene, styrene, phenylacetylene, benzene and toluene have then been investigated by ^1H nuclear magnetic resonance (NMR) (Supplementary Figure 40-47). Interestingly, **DPA[3]** experienced small changes in their ^1H NMR chemical shifts when adding guest molecules of alkenes and alkynes like *n*-hexylene, *n*-hexyne, styrene and phenylacetylene, implying the existence of host-guest interaction, perhaps weak CH/ π interaction or electrostatic interaction, between **DPA[3]** and these molecules. In comparison, the unchanged ^1H NMR chemical shifts of **DPA[3]** suggested that no host-guest interaction occurs between **DPA[3]** and other small molecules (e.g. *n*-hexane, ethylbenzene, benzene

and toluene). Therefore, the above results might disclose the underlying relationship between the observed edge to face packing in the solid state of **DPA[3]** and its host-guest interaction with ethene and ethyne.

Some discussions have been presented in the revised manuscript:

*“However, the interaction between DPA[3] and guest molecules is uncertain because ethene and ethyne exhibit disorder in the crystal of DPA[3]⊃ethene and DPA[3]⊃ethyne. DFT calculations implied that DPA[3] favors binding to π -bonded ethene and ethyne rather than ethane which is a saturated hydrocarbon (Supplementary Figure 39). Additionally, ^1H NMR investigations of host-guest complexes between DPA[3] and a series of small molecules indicated the existence of weak host-guest interaction between DPA[3] and alkenes and alkynes such as *n*-hexylene, *n*-hexyne, styrene and phenylacetylene (Supplementary Figures 40-47). Thus, we infer that the formation of host-guest complex might be attributed to the weak CH- π or electrostatic interaction between DPA[3] and guest⁶⁰⁻⁶².”*

However, this work is interesting, (as is the direct observation of included C₂H₂ and C₂H₄ by ^1H NMR spectroscopy), and thus deserves publication in large audience journals, such as Chemistry, a European Journal, or Communications Chemistry.

Reply: We appreciate the reviewer’s interest in our work and thank you for your comments and thoughtful suggestions that clearly improved the quality of our manuscript. With these changes and responses, we hope the revised manuscript is now acceptable for publication in Nature Communications.

REVIEWERS' COMMENTS

Reviewer #1 (Remarks to the Author):

The authors have fully addressed my previous questions. In addition, they have included new experimental results which fully addressed the main concerns of the other referee and make stronger the conclusions of this work. Therefore, I suggest the publication of this manuscript in its present form.

Reviewer #2 (Remarks to the Author):

The revised version of this manuscript includes new experiments, addressing the main comments raised during the first round. These experiments include first the synthesis of new DPAs derivatives, to assess the origin of the stacking effects observed in the solid state. The revised version also includes the study of the hydrodynamic behavior of some DPAs by DOSY NMR experiments. These experiments are convincing, and reinforce the conclusions of the authors.

I acknowledge the fact that the authors have now significantly reinforced the conclusions of their manuscript.

I thus consider this paper as suitable for publication in Nature Communications.

Minor point: line 294, "the decrease of the hydrodynamic size" should be replaced by "the increase of the hydrodynamic size"

REVIEWER COMMENTS

Reviewer #1 (Remarks to the Author):

The authors have fully addressed my previous questions. In addition, they have included new experimental results which fully addressed the main concerns of the other referee and make stronger the conclusions of this work. Therefore, I suggest the publication of this manuscript in its present form.

Reply: We greatly appreciate the reviewer's positive comments on the work in this manuscript.

Reviewer #2 (Remarks to the Author):

The revised version of this manuscript includes new experiments, addressing the main comments raised during the first round. These experiments include first the synthesis of new DPAs derivatives, to assess the origin of the stacking effects observed in the solid state. The revised version also includes the study of the hydrodynamic behavior of some DPAs by DOSY NMR experiments. These experiments are convincing, and reinforce the conclusions of the authors.

I acknowledge the fact that the authors have now significantly reinforced the conclusions of their manuscript.

I thus consider this paper as suitable for publication in Nature Communications.

Reply: We greatly appreciate the reviewer's positive comments on the work in this manuscript.

Minor point: line 294, "the decrease of the hydrodynamic size" should be replaced by "the increase of the hydrodynamic size"

Reply: We have replaced "the decrease of the hydrodynamic size" with "the increase of the hydrodynamic size".